# Predictive role of ctDNA in esophageal squamous cell carcinoma receiving definitive chemoradiotherapy combined with toripalimab

Baoqing Chen [1,2,8], Shiliang Liu [1,2,8], Yujia Zhu [1,2,8], Ruixi Wang[1,2,8], Xingyuan Cheng[1,2], Biqi Chen[1,2], Mihnea P. Dragomir[3,4,5], Yaru Zhang[6], Yonghong Hu[1,2], Mengzhong Liu [1,2], Qiaoqiao Li [1,2,9] ✉, Hong Yang [1,7,9] ✉ & Mian Xi [1,2,9] ✉

The combination of toripalimab (an anti-PD-1 antibody) with definitive chemoradiotherapy (CRT) demonstrated encouraging efficacy against locally advanced esophageal squamous cell carcinoma (ESCC) in the EC-CRT-001 phase II trial (NCT04005170). The primary endpoint of this trial was the clinical complete response rate (cCR), and the secondary endpoints included overall survival (OS), progression-free survival (PFS), duration of response, and quality of life. The exploratory analyses of EC-CRT-001 include exploring the role of circulating tumor DNA (ctDNA) and blood-based tumor mutational burden (bTMB) in predicting the response and survival. In total, 118 blood and 35 tissue samples from 42 enrolled patients were included in the analyses. We found that ctDNA-negative patients achieved a higher cCR compared to those with detectable ctDNA during CRT (83%, 19/23 vs. 39%, 7/18; $p = 0.008$) or post-CRT (78%, 21/27 vs. 30%, 3/10; $p = 0.017$). Patients with detectable ctDNA during CRT had shorter PFS ($p = 0.014$). Similarly, patients with post-CRT detectable ctDNA had a significantly shorter PFS ($p = 0.012$) and worse OS ($p = 0.004$). Moreover, patients with high bTMB levels during CRT had prolonged OS ($p = 0.027$). In conclusion, ctDNA and bTMB have the potential to predict treatment efficacy and survival in ESCC treated with CRT and immunotherapy.

Esophageal carcinoma is one of the leading causes of cancer-related deaths worldwide. Esophageal squamous cell carcinoma (ESCC) constitutes more than 80% of all esophageal carcinoma cases and is largely prevalent in less developed countries[1]. Definitive chemoradiotherapy (CRT) is recommended for patients with locally advanced ESCC who are ineligible for surgery. Although long-term survival and survival benefits comparable to surgery have been achieved, clinical trial data still show a 3-year overall survival (OS) rate of <50% after definitive CRT, indicating that more effective regimens are urgently needed[2,3].

Considering the improved survival observed in recent randomized phase III trials, PD-1 inhibitors combined with chemotherapy are recommended as first-line treatment approaches for advanced ESCC[4,5]. The combination of PD-1 inhibitors and CRT in locally advanced ESCC is currently being investigated in several clinical trials[6,7]. EC-CRT-001(NCT04005170) is the phase II trial that explored the efficacy and safety of combining toripalimab, an anti-PD-1 monoclonal antibody, with definitive CRT in locally advanced ESCC, which showed promising clinical efficacy with acceptable toxicities[8]. However, approximately

40% of patients had residual disease and progressed after a certain time after this combination treatment. Therefore, the discovery of effective biomarkers for identifying patients who will achieve long-lasting responses to CRT in combination with immunotherapy is crucial.

Liquid biopsy, especially the detection of circulating tumor DNA (ctDNA) from peripheral blood, is a promising non-invasive approach to monitor tumor response and progression[9]. The role of ctDNA as a biomarker to detect minimal residual disease (MRD) and for the guidance of adjuvant treatment after curative surgery has been widely accepted[10,11]. Recently, studies focusing on the detection of MRD by ctDNA after completion of definitive CRT with curative intent have also been performed for lung cancer and esophageal carcinoma[12,13]. The application of next-generation sequencing (NGS) for detecting ctDNA mutations, which are released by tumor cells into the bloodstream, has been increasingly utilized. However, the application of NGS to monitor ctDNA and correlate it to response and survival in patients with ESCC receiving CRT in combination with immunotherapy has not been explored. Tumor mutational burden (TMB) refers to the number of acquired somatic mutations per megabase of the tumor genome[14] NGS-based estimation of tissue TMB has been widely used in clinical settings as a predictive biomarker for immunotherapy, but results have been inconsistent[15]. The inconsistency may be due to sampling bias and tumor heterogeneity in biopsies obtained from a single site. Alternatively, blood-based TMB (bTMB) determined by NGS analysis of ctDNA offers a non-invasive and convenient alternative to tissue TMB. BTMB has been shown to have comparable efficacy to tissue TMB as a surrogate biomarker for representing immunogenicity and predicting the response to immunotherapy in lung cancer[16]. However, the role of bTMB in ESCC remains unclear.

This exploratory analysis of EC-CRT-001 was to investigate whether ctDNA and bTMB could predict the response and recurrence in patients with localized ESCC treated with definitive CRT combined with toripalimab.

## Results

### Patient characteristics and outcome

The baseline clinical characteristics were shown in Supplementary Table 1. The patients were predominantly men (76%, $n = 32$). Of the 42 patients, 62% and 38% were diagnosed with stages I-III and IVA disease, respectively. Three months after CRT, a promising clinical complete response (cCR) rate of 62% (95% CI, 46–76) was achieved in 26 patients. With a median follow-up of 27.6 months (95% CI, 25.2–30.0), the median progression-free survival (PFS) of the whole cohort was 12.2 months (95% CI, 8.4–16.0), while the median OS was not reached.

Figure 1A had shown the workflow of this study. After excluding two patients without available baseline biopsy specimens and five patients with tissue specimens that failed in quality control, tissues from 35 of 42 patients were eligible for baseline genomic analyses. After excluding two patients without pretreatment plasma samples, the ctDNA data of the remaining 40 patients were eligible for baseline analyses. For ongoing and post-CRT ctDNA and bTMB analyses, 41 patients in the third week of CRT and 37 patients after CRT had plasma available for ctDNA and bTMB analyses. Dynamic ctDNA analyses were performed for 36 patients. We profiled 118 blood and 35 tissue samples from 42 enrolled patients. The panel covering 474 cancer-relevant genes was applied (Supplementary Fig. 1).

### Profiling of baseline ctDNA

Baseline ctDNA analysis revealed 168 somatic mutations in 40 patients, with a median of four mutations per patient (range, 1–13 mutations). The most frequently mutated genes were *TP53* (68%, $n = 27$), *CDKN2A* (20%, $n = 8$), *NFE2L2* (15%, $n = 6$), and *LRP1B* (13%, $n = 5$) (Fig. 1B). Tumor somatic mutations and aligned ctDNA analyses revealed 147 somatic variants in 33 patients. Plasma and tissue shared 85 (58%) mutations,

whereas 62 (42%) of the mutations were unique to plasma ctDNA analyses, suggesting that plasma can be a supplement to tumor tissue samples or even enriched in clones with metastatic potential. The most frequently mutated genes in the plasma were also observed in tissues, including *TP53*, *CDKN2A*, and *NFE2L2* (Fig. 1C, D). Missense variants were the most frequently detected in both tissues and plasma (Supplementary Fig. 2). The VAF of the dominant clone mutation in the tumor tissue was positively correlated with its corresponding baseline plasma mutation, indicating a positive correlation between the abundance of mutations in ctDNA and tumor DNA (r = 0.447, $p < 0.001$) (Supplementary Fig. 3A). The allele frequency (AF) of shared variants in the tumor or plasma was significantly higher than that of the unique mutations (Supplementary Fig. 3B). Only NFE2L2 mutation was negatively correlated to with PFS (HR, 3.33; 95% CI, 1.28–8.63; $p = 0.009$) and OS (HR, 3.93; 95% CI, 1.47–10.5; $p = 0.003$) in the univariate analyses (Supplementary Fig. 4 and Supplementary Table 2). KEGG pathway analyses also demonstrated that the NFE2L2 pathway was correlated with shorter survival (PFS: HR, 3.24; 95% CI, 1.30–8.08; $p = 0.008$) and OS (HR, 3.85; 95% CI, 1.50–9.91; $p = 0.003$) (Supplementary Table 3 and Supplementary Fig. 5).

CtDNA was detected in 29 (73%) of the 40 patients at baseline. In those patients with detectable ctDNA, a median of six mutations (range, 1–13) were detected and median absolute cell-free DNA concentration was 26.3 ng/mL (range, 9.6–52.9). Longer tumor diameter ($p = 0.023$) and advanced stage ($p = 0.036$) were associated with higher ctDNA-positivity rates before treatment (Fig. 1E and Supplementary Table 4). Maximal tumor somatic VAF (maxVAF) in the plasma represents the largest mutated ctDNA clone and is an indicator of the overall ctDNA quantity. Similar to the ctDNA-positivity rate, longer tumor diameter ($p = 0.011$) and stage IVA disease ($p = 0.037$) were correlated with a higher maxVAF (Fig. 1F).

### Dynamic ctDNA profiling is associated with clinical response

Figure 2A had shown the ctDNA landscapes during and after CRT. The ctDNA positivity rate decreased from 73% (29/40) at baseline to 44% (18/41) during CRT, and further to 27% (10/37) after CRT (Fig. 2B). Similarly, the mean maxVAF in ctDNA decreased from 2.25% at baseline to 0.33% during CRT and to 0.27% after CRT, suggesting that the overall tumor burden continued to decline during CRT (Fig. 2C).

There was no significant difference in the cCR rates between ctDNA-negative (73%, 8/11) and ctDNA-positive (59%, 17/29) patients at baseline ($p = 0.486$). Compared to those with detectable ctDNA, a higher cCR rate was observed in patients without detectable ctDNA during CRT (83%, 19/23 vs. 39%, 7/18; $p = 0.008$). Patients with detectable ctDNA levels after CRT showed a significantly poorer response to therapy, wherein only 30% (3/10) of patients achieved cCR. In contrast, 78% (21/27) of the patients with negative cDNA results achieved cCR ($p = 0.017$) (Fig. 2D).

We then analyzed the correlation between dynamic changes in ctDNA status and the therapy response of 36 patients who had plasma samples collected at three time points (baseline, during, and after CRT). We categorized the change in ctDNA during treatment into four patterns: pattern 1 (continuously ctDNA-negative at all three time points, $n = 7$); pattern 2 (ctDNA cleared during CRT, $n = 12$,), pattern 3 (ctDNA cleared after CRT, $n = 8$), and pattern 4 (ctDNA still positive after CRT, $n = 9$), with corresponding CR rates of 100% (7/7), 83% (10/12), 50% (4/8), and 33% (3/9), respectively (Fig. 2E). Compared to patients with pattern 4, a higher cCR rate was found in patients with continuously negative ctDNA (pattern 1; $p = 0.011$) and those with rapid ctDNA clearance during CRT (pattern 2; $p = 0.032$).

### During and post-CRT ctDNA and its dynamic change patterns indicate their potential as biomarkers for survival

We evaluated the prognostic value of dynamic ctDNA by comparing the survival differences between ctDNA-positive and -negative patients

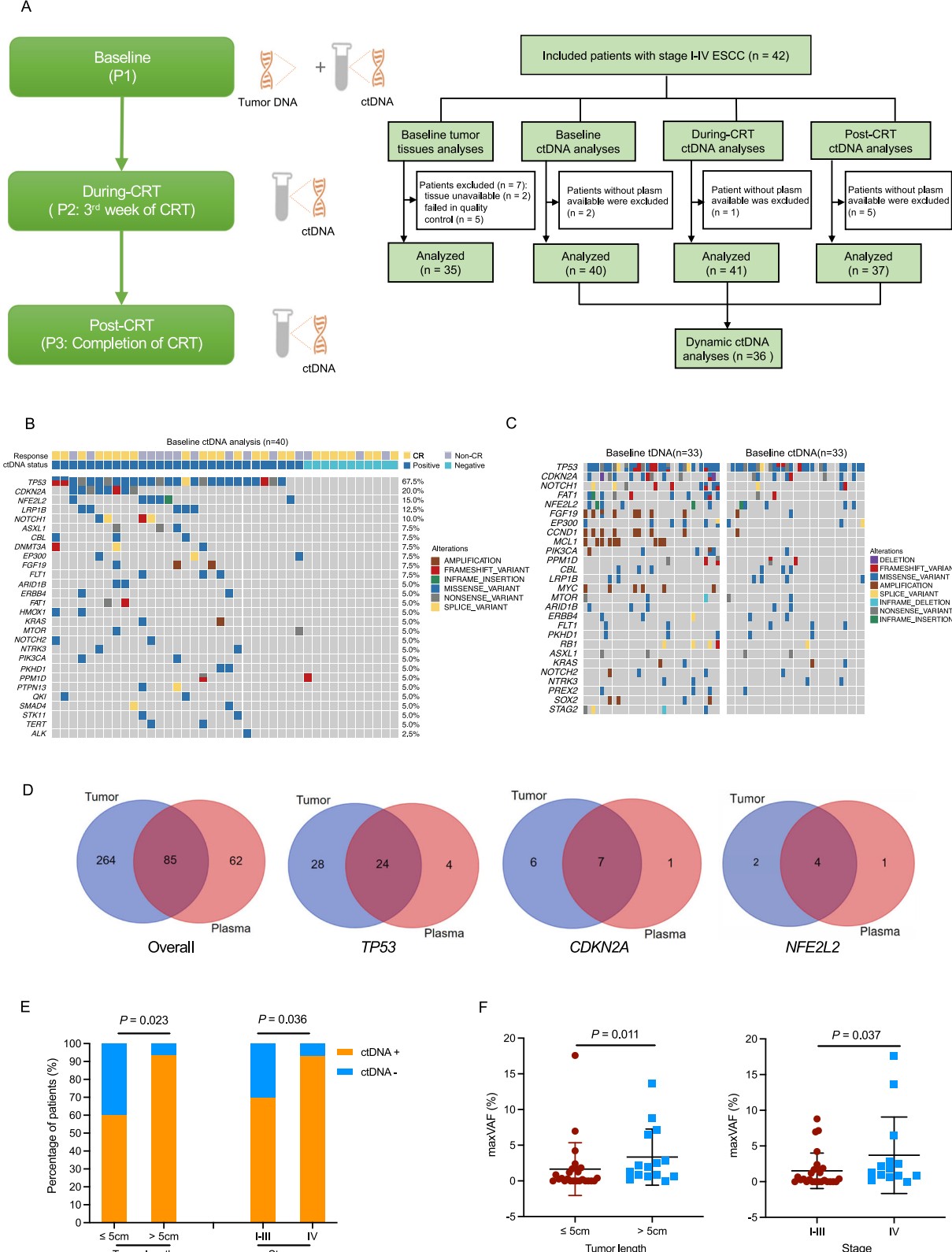

**Fig. 1 | Pre-CRT tumor and ctDNA analyses in patients with locally advanced ESCC. A** The workflow of the study; **B** Distribution of genetic variation in ctDNA genotyping analyses: each column represents an individual patient. Clinical characteristic of response and stage are shown at the top. Genes mutated in at least 2.5% of the patients in our cohort are depicted. The fraction of tumors with mutations in each gene is shown on the left; **C** Genotyping results of 33 patients with both tumor tissue and ctDNA analyses; **D** Venna graph present the number

sharing variant of all, *TP53*, *CDKN2A*, and *NFE2L2* in plasms and tissue. Comparison of ctDNA positive rate (**E**) and mean maxVAF (**F**) in subgroups according to tumor length (*n* = 42) and disease stage (*n* = 42). Data are presented as mean values +/− SD. *P*-values (*p*) were determined by two-tailed unpaired t-tests. ESCC esophageal squamous cell carcinoma, CRT chemoradiotherapy, CR complete response, ctDNA circulating tumor DNA, maxVAF maximal variant allele frequency. Source data are provided as a Source Data file.

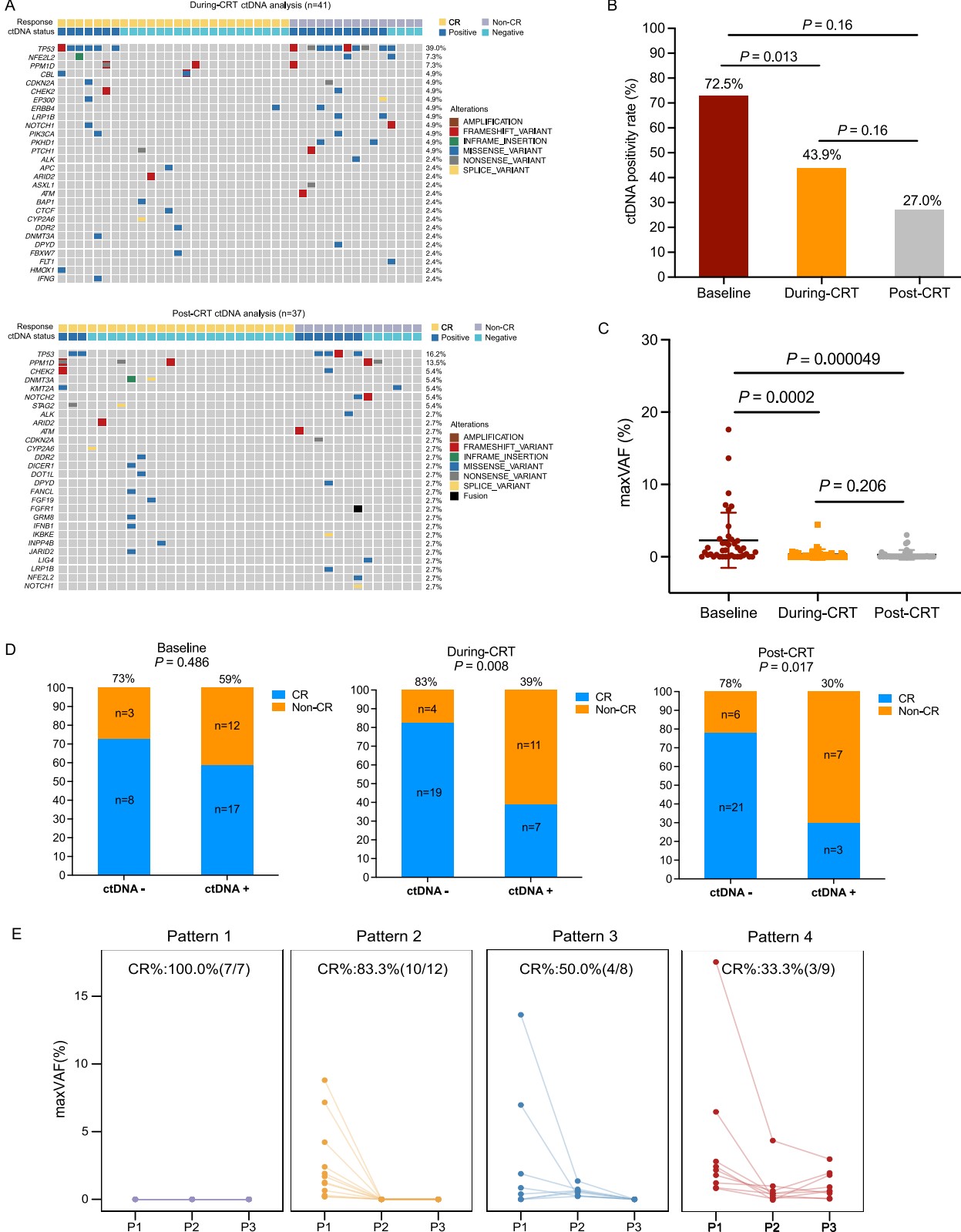

**Fig. 2 | Dynamic ctDNA profiling are correlated with tumor response.**
**A** Distribution of genetic variation in during- and post-CRT ctDNA genotyping analyses; CtDNA-positive rate (%) (**B**) and maxVAF (%) (**C**) decrease during (*n* = 41) and after (*n* = 37) CRT in comparison to baseline (*n* = 40). Data are presented as mean values +/− SD. *P*-values (*p*) were determined by two-tailed Fisher's exact test (**B**) and two-tailed unpaired t-tests (**C**); **D** Difference in cCR rate among ctDNA-positive and ctDNA-negative groups at baseline, during, and after CRT. *P*-values

(*p*) were determined by two-tailed Fisher's exact test; **E** Difference in cCR rate in subgroups categorized by the ctDNA change patterns during CRT (pattern 1, ctDNA continuously negative at all three time points, *n* = 7); pattern 2 (ctDNA cleared during CRT, *n* = 12), pattern 3 (ctDNA cleared after CRT, *n* = 8), and pattern 4 (ctDNA was still positive after CRT, *n* = 9). cCR clinical complete response, ctDNA circulating tumor DNA, maxVAF maximal variant allele frequency, CRT chemoradiotherapy. Source data are provided as a Source Data file.

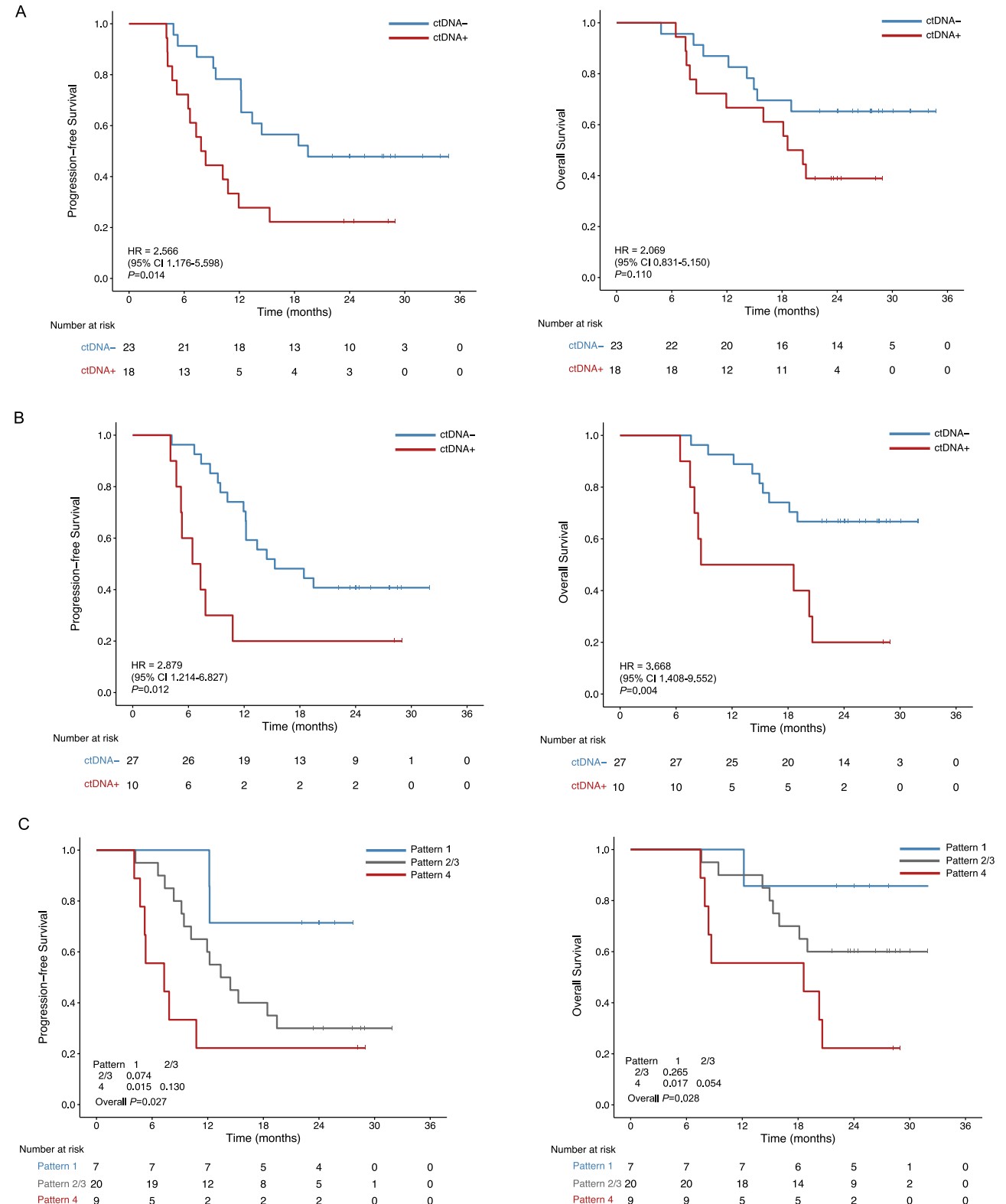

**Fig. 3 | During and post-CRT ctDNA status and its dynamic change patterns predict survival.** Progression-free and overall survival of ctDNA-positive and ctDNA-negative patients during CRT (**A**) and post-CRT (**B**) and in patients with different dynamic change patterns (**C**). *P*-values (*p*) were determined by Log Rank Test. *ctDNA* circulating tumor DNA, *CRT* chemoradiotherapy. Source data are provided as a Source Data file.

at different time points. Among ctDNA-positive and -negative patients at baseline, there was no significant difference in PFS (HR, 1.50; 95% CI, 0.60–3.73; *p* = 0.384) or OS (HR, 2.35; 95% CI, 0.68–8.09; *p* = 0.162) (Supplementary Fig. 6). Patients with detectable ctDNA during CRT had significantly shorter PFS compared to ctDNA-negative patients (HR, 2.57; 95% CI, 1.18–5.60; *p* = 0.014), while OS was comparable between these two groups (HR, 2.07; 95% CI, 0.83–5.15; *p* = 0.110) (Fig. 3A). Patients with detectable ctDNA post-CRT had also a

**Table 1 | Sensitivity, specificity, PPV, and NPV of ctDNA and PET-CT in predicting cCR**

| | Sensitivity | p-value[a] | Specificity | p-value[a] | PPV | p-value[a] | NPV | p-value[a] |
|---|---|---|---|---|---|---|---|---|
| PET-CT at three months after CRT | 83% (19/23) | – | 100% (11/11) | – | 100% (19/19) | – | 73% (11/15) | – |
| During-CRT ctDNA | 70% (16/23) | 0.49 | 82% (9/11) | 0.48 | 89% (16/18) | 0.23 | 56% (9/16) | 0.46 |
| Post-CRT ctDNA | 87% (20/23) | 0.99 | 64% (7/11) | 0.09 | 83% (20/24) | 0.12 | 70% (7/10) | 0.99 |

*P*-values (*p*) were determined by two-tailed Fisher's exact test. Source data are provided as a Source Data file.
*cCR* clinical complete response, *PPV* positive predictive value, *NPV* negative predictive value.
[a]Compared to PET-CT at three months after CRT.

significantly increased risk of disease progression (HR, 2.88; 95% CI, 1.21–6.83; *p* = 0.012) and death (HR, 3.67; 95% CI, 1.41–9.55; *p* = 0.004) (Fig. 3B). Higher post-treatment maxVAF was correlated to poor PFS (HR, 1.94; 95% CI, 1.16–3.27; *p* = 0.011). No other significant correlations were found between the maxVAF of ctDNA and survival (Supplementary Table 5).

Next, we investigated whether the patterns of ctDNA changes predicted survival by comparing patients with continually positive ctDNA (pattern 1) to those whose ctDNA was cleared during or after CRT (pattern 2/3). No significant difference of PFS (HR, 0.38; 95% CI, 0.13–1.10; *p* = 0.074) and OS (HR, 0.44; 95% CI, 0.10–1.88; *p* = 0.265) was observed. The risk of recurrence and mortality rate showed a statistically significant increase in patients with uncleared ctDNA after CRT compared with those with continuously negative ctDNA (pattern 1 vs. 4; HR, 0.44; 95% CI, 0.044–0.72; *p* = 0.015 for PFS; HR, 0.18; 95% CI, 0.044–0.73; *p* = 0.017 for OS) (Fig. 3C).

PET-CT is one of the pivotal approaches to evaluate patient response after completion of CRT. In our cohort, residual disease was evaluated in 38 patients using PET-CT performed three months after CRT. Among 34 patients who underwent both during-ctDNA, post-ctDNA, and PET-CT assessments, PET-CT yielded a sensitivity of 83% (19/23) and specificity of 100% (11/11) for cCR prediction. CtDNA achieved a sensitivity of 70% (16/23) and specificity of 82% (9/11) during CRT, and a sensitivity of 87% (20/23) and specificity of 64% (7/11) after CRT for cCR prediction. No significant differences in sensitivity and specificity were found between ctDNA and PET-CT (Table 1). The survival rates of patients with non-cCR confirmed by PET-CT and those with detectable ctDNA after CRT were comparable (Fig. 4A).

Furthermore, ctDNA seems to be an effective supplementary method to PET-CT for differentiating radiation esophagitis from other residual diseases. Figure 4B illustrates one (ID 17) of the two patients with PET-avid lesions after CRT with a maximum standardized uptake value (SUVmax) of 5.2 and 5.8, overlapping the primary esophageal tumor. Radiation esophagitis or residual disease should be differentially diagnosed in these patients. Considering these two cases, both had negative ctDNA levels after CRT, and radiation esophagitis was diagnosed instead of residual disease. This conclusion was further confirmed by the long PFS observed in these two patients at 28.9 and 28.5 months, respectively.

### Higher bTMB during or post-CRT predicts better survival

We further explored whether bTMB could be used as an alternative biomarker to predict response and survival. The median bTMB values were 3 (IQR, 1–7), 1 (IQR, 1–3), and 1 (IQR, 1–3) at baseline, during CRT, and post CRT, respectively. The relationship between clinical outcomes and bTMB scores was evaluated by determining the bTMB scores for all available samples. No significant correlation was found between PD-L1 combined positive score (CPS) (r = 0.079, *p* = 0.66) or tissue TMB (r = 0.099, *p* = 0.55) with bTMB. No difference in survival was found between the bTMB-high and -low groups at baseline at the different cut-off points (1, 2, 3, and 4). The forest plots comparing bTMB-high or -low subgroups during or post-CRT at different cut-off values showing HRs and 95% CI of the PFS and OS are presented in Fig. 5A, B. Patients with high bTMB (>1 mutations/mb) detected during

CRT were associated with a favorable OS (HR, 0.33; 95% CI, 0.13–0.88; log-rank *p* = 0.027) and a non-significant trend towards better PFS (HR, 0.49; 95% CI, 0.22–1.08; *p* = 0.077) (Fig. 5C). Patients with higher bTMB (>3 mutations/mb) detected post-CRT were associated with improved PFS (HR, 0.28; 95% CI, 0.08–0.96; *p* = 0.03) but not OS (HR, 0.46; 95% CI, 0.13–1.60, *p* = 0.21) (Supplementary Fig. 7). The bTMB-high and -low subgroups showed no significant differences in cCR rates during or after CRT (Supplementary Fig. 8).

### Discussion

This study explores the role of ctDNA and bTMB in monitoring the response and survival to CRT combined with immunotherapy in ESCC. Our findings demonstrate the predictive value of ctDNA dynamic analyses and the use of bTMB to predict the clinical benefits for patients with ESCC undergoing anti-PD1 immunotherapy plus CRT.

It has been established that radiation-induced cell death leads to the release of tumor neoantigens and proinflammatory factors and promotes an antitumor immune response. A similar and synergistic antitumor effect was also observed when anti-PD-1 treatment was combined with radiation, leading to a satisfactory pathologic complete response in ESCC when PD-1 inhibitors were combined with neoadjuvant CRT[17,18]. Toripalimab combined with CRT achieved promising efficacy in this phase II trial, which showed a cCR of 62%, providing evidence of this treatment approach for locally advanced ESCC[8]. However, the lack of correlation between treatment response or survival of PD-L1 expression and biomarkers, such as tissue TMB, limited their predictive ability. The failure of tissue TMB in predicting response and survival may be explained by molecular variability and genetic diversity due to intratumoral heterogeneity, warranting the need for novel biomarkers to predict patient response to trimodal approaches and further identify patients at risk of tumor progression. Liquid biopsy-based biomarkers are potential therapeutic candidates[19,20].

First, we explored whether baseline ctDNA levels could be an indicator of the tumor burden. Recently, the prediction of treatment response and surveillance using ctDNA has been widely explored. Nevertheless, the ctDNA status can vary depending on the cancer type and disease stage. For example, the ctDNA positivity rate increased in concordance with that in esophageal cancer[21] In addition, the ctDNA positivity rate was positively correlated with advanced stage, tumor size, and regional metastatic disease among patients with inoperable lung cancer who received CRT. However, no correlation was found between patient survival and baseline ctDNA status[19]. In the current study, we observed similar results, wherein ctDNA was more frequently detected in patients with stage IVA disease and longer tumor diameter. Collectively, these findings suggest that baseline ctDNA may serve as a biomarker of tumor burden and encouraged us to explore the prognostic power of ctDNA.

Hence, we secondly clarified the role of ctDNA in predicting the prognosis of patients with localized ESCC. The short half-life (a few hours) of ctDNA enables it accurately, reflecting tumor burden in real time, particularly during treatment, where dynamic ctDNA status robustly correlates with dynamic tumor burden, providing a strong rationale for using dynamic ctDNA to monitor treatment response[20]. The application of ctDNA was confirmed in patients with rectal cancer

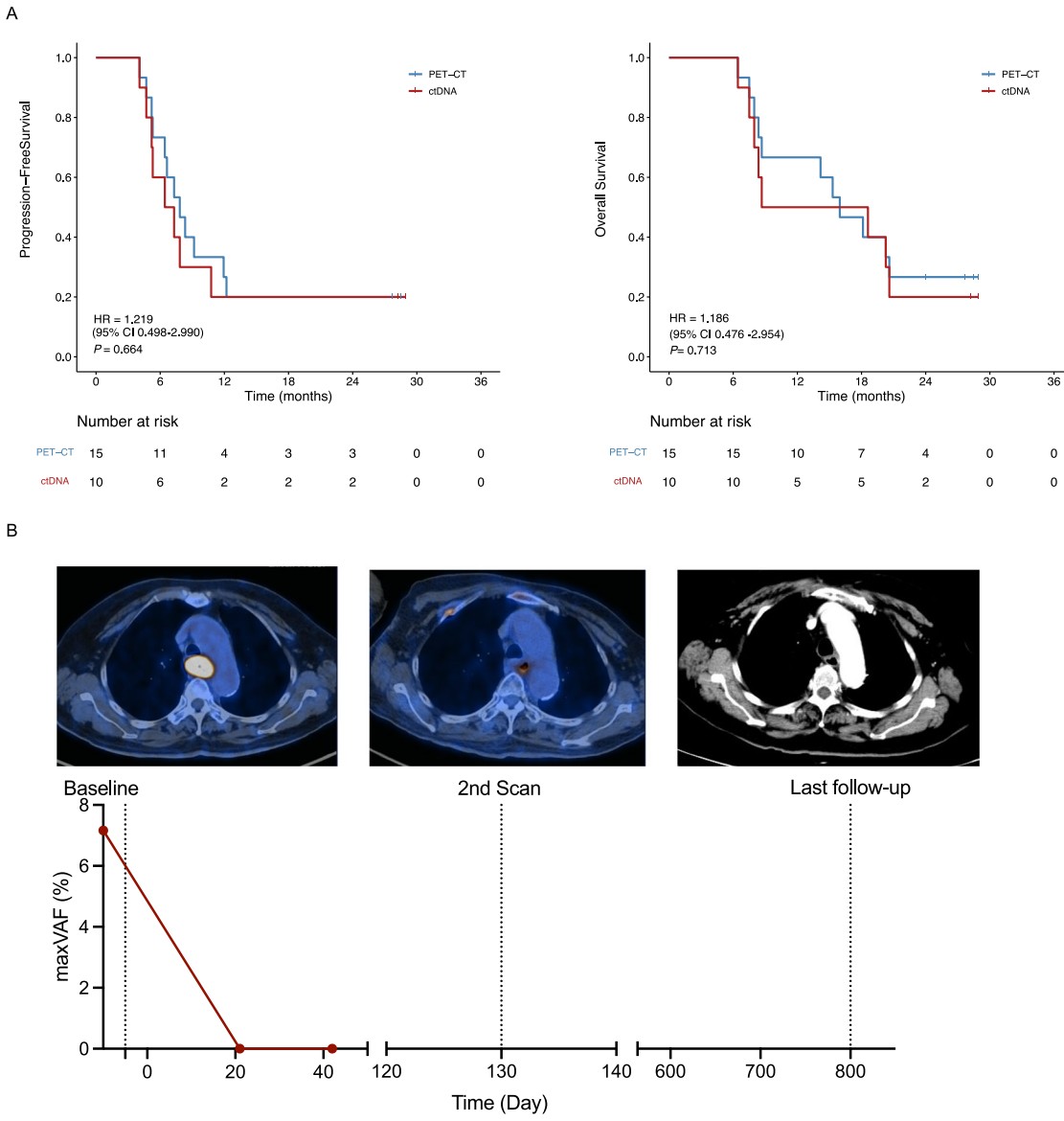

**Fig. 4 | ctDNA status predicts progression-free survival and overall survival similar to PET-CT imaging. A** PFS and OS of patients with detectable ctDNA after CRT or non-cCR patients confirmed by PET-CT at 3 months after CRT. *P*-values (*p*) were determined by Log Rank Test; **B** Patient (#ID17) with stage IIIA ESCC was ctDNA-negative after CRT but the PET-CT imaging at 3 months after CRT had shown an SUVmax of 5.8 on the site of the primary esophageal lesion. This patient was confirmed to have radiation esophagitis rather than residual disease by the long PFS. ESCC esophageal squamous cell carcinoma, ctDNA circulating tumor DNA, PFS progression-free survival, OS overall survival, maxVAF maximal variant allele frequency, CRT chemoradiotherapy. Source data are provided as a Source Data file.

where ctDNA-positive rates during post-neoadjuvant CRT were significantly higher among poor responders[22,23]. Similar results have been reported in lung cancer[19] The correlation between ctDNA changes after the first cycle of chemotherapy and later response in advanced ESCC resonates with the findings of our study, which showed that ctDNA detected during and after post-CRT, rather than baseline ctDNA, correlated with an unfavorable treatment response[24]. In addition, ctDNA clearance during CRT predicts a higher possibility of complete remission after CRT. In concordance, patients with esophageal adenocarcinoma who received post-neoadjuvant CRT, underwent surgery, and have detectable ctDNA during surveillance have a higher likelihood of recurrence compared to those with persistent negative ctDNA[25]. In the setting of definitive CRT, a similar result was also reported in esophageal adenocarcinoma[13]. These findings support the use of dynamic ctDNA changes, rather than baseline ctDNA, to predict short-term treatment response and monitor relapse and survival in ESCC.

Third, we confirmed the predictive role of bTMB for survival. Anti-PD-1/PD-L1 immunotherapy was recently approved as one of the first-line approaches for esophageal carcinoma, but the poor response rate of 10–30% in patients with ESCC indicates that only a small group of patients will benefit from anti-PD-1/PD-L1[26]. The inconsistent predictive value of candidate biomarkers, PD-L1 expression, tissue TMB, and the lack of serial tumor tissue for molecular and genomic profiling limit the application of PD-L1 expression and tissue TMB[26]. Here, we found that ctDNA could also be used in predicting the response and survival of patients with ESCC treated with anti-PD-1 antibody when combined with CRT. Moreover, the quantified ctDNA status can be supplemented with bTMB data, which is another attractive biomarker. Post-hoc analyses of two large randomized trials validated significantly prolonged PFS and higher bTMB in patients with lung cancer undergoing nivolumab treatment[27]. The commonly reported cut-off value for high TMB, which is typically set at 16, has been observed mainly in studies focusing on lung cancer, where TMB tends to be higher compared to

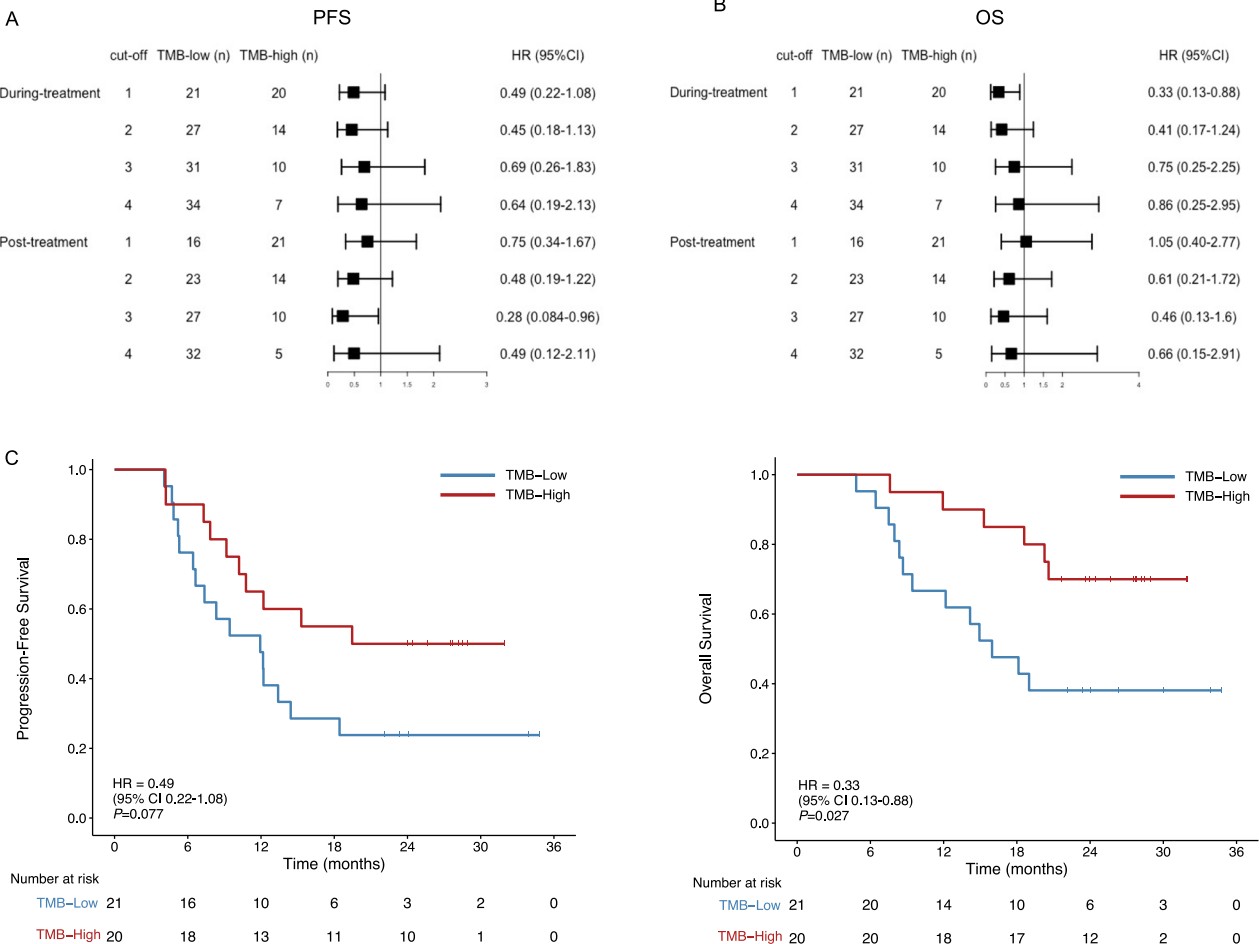

**Fig. 5 | Survival of bTMB-high and bTMB-low groups.** Forest plots show HRs and 95% CI of the PFS (**A**) and OS (**B**) in bTMB-high or -low subgroups divided by during- or post-CRT bTMB according to cut-off values at 1, 2, 3, and 4. Data are presented as the hazard ratios (HR) with error bars showing 95% confidence intervals. **C** PFS and OS of patients with high bTMB (>1.0 mutations/mb) and low bTMB (≤1.0 mutations/mb) detected during CRT. *P*-values (*p*) were determined by Log Rank Test. HR hazard ration, CI confidence intervals, OS overall survival, bTMB blood-based tumor mutational burden, CRT chemoradiotherapy. Source data are provided as a Source Data file.

other cancer types[28]. However, in a report involving 1145 ESCC patients, the median tissue TMB value was found to be 3.58, which is significantly lower than the values observed in melanoma and lung cancer[29]. In our study, we observed lower median bTMB values, which could be attributed to the localized stage of the cancer and its specific type. It is crucial to consider the context of different cancer types, stages, and treatment approaches when interpreting TMB values and establishing the optimal cut-offs for clinical significance.

Additionally, we revealed the value of ctDNA in distinguishing patients with residual disease. The significance of MRD is in identifying individuals with residual disease after curative intent treatment and those who may benefit immensely from adjuvant treatment. The identification of MRD in such individuals using ctDNA has been widely reported for many localized cancers after radical surgery. The value and accuracy of ctDNA in guiding adjuvant treatment for stage II colon cancer and urothelial carcinoma after surgery have recently been validated in robust phase III clinical trials[30,31]. In some locally advanced cancers, CRT is a definitive treatment with tumor control efficacy comparable to that of surgery. Preliminary investigations to determine MRD in esophageal cancer after CRT have yielded results but thus so far, the role of ctDNA for the detection of MRD has not been validated in prospective studies, which are needed to guide adjuvant treatment in ESCC[13]. Additionally, we showed the potential role of ctDNA in discriminating patients with residual disease from those with esophagitis.

We envision the use of ctDNA combined with PET-CT not only for conventional response evaluation but also for differential diagnosis between residual disease and non-malignant inflammation, including radiation-induced esophagitis. Comparable survival rates between non-cCR patients and those with detectable ctDNA after CRT suggest that post-CRT ctDNA may have a predictive value similar to that of PET-CT at earlier time points. However, due to the limited number of patients enrolled, we acknowledge that our study may lack a sufficient statistical power to draw a definitive conclusion on the role of ctDNA as an adjunct to PET-CT. Further prospective research with larger samples comparing these two modalities could offer further evidence in this regard.

This study has several limitations. First, the modest sample size warrants the need for highly powered studies to validate the use of ctDNA as a predictive tool for anti-PD-1 treatment combined with CRT in ESCC. Second, since no commercial NGS platform with a standardized pipeline was available for ESCC, the clinical validity and utility of the customized NGS panel used in the current study need to be investigated. Third, low mVAF during and post-CRT indicated the detection of a limited quantity of ctDNA, which may have underestimated bTMB. Low level of bTMB was detected in this study, which also revealed this potential underestimation due to the low shedding of ctDNA.

In conclusion, we report that ctDNA negativity and higher bTMB levels correlated with better tumor response and survival in patients

with ESCC who underwent CRT combined with toripalimab. In the future, we aim to use these two biomarkers to distinguish patients at higher risk of relapse and to determine who will benefit from subsequent intense adjuvant therapy.

## Methods

### Patient enrollment, treatment, and surveillance

This single-arm, phase II trial (EC-CRT-001) was performed at the Sun Yat-sen University Cancer Center to explore the efficacy and safety profile of toripalimab in combination with definitive CRT in patients with unresectable locally advanced ESCC. Detailed information on participants and study design has been previously reported[8]. Briefly, from November 2019 to January 2021, 42 patients were treated with concurrent thoracic radiotherapy, chemotherapy, and toripalimab, followed by maintenance treatment with toripalimab. Radiotherapy was administered by applying intensity-modulated radiotherapy 5 days per week (days 1 to 38) at a total dose of 50.4 Gy with 28 fractions. Concurrently, a total of five cycles of paclitaxel (50 mg/m$^2$) plus cisplatin (25 mg/m$^2$) were weekly administered. During CRT, the patients received two cycles of intravenous toripalimab (240 mg) on days 1 and 22. After the completion of concurrent CRT, intravenous toripalimab (240 mg every three weeks) was continued for up to 1 year, or to disease progression or with unacceptable toxicity. The primary endpoint was the cCR rate at 3 months after the completion of radiotherapy. Tumor response was assessed according to the Response Evaluation Criteria in Solid Tumors (RECIST; version 1.1) using CT, PET, and esophagogastroduodenoscopy. CR determined by PET was defined when the SUVmax in the primary region was at a normal physiological level or when SUVmax was higher than normal but exhibited a uniform distribution and overlapped with the radiation field, indicating esophagitis[32]. Secondary endpoints included OS and PFS. Exploratory outcomes included associations of clinical response and survival with ctDNA status, genetic biomarkers, and TMB. The study was conducted in accordance with the international standards of the Declaration of Helsinki and Good Clinical Practice. The Institutional Review Board of Sun Yat-sen University Cancer Center approved this study and informed consent was obtained from all participants before enrollment.

### Tumor biopsies and blood collection

Of the 42 enrolled patients, baseline biopsy specimens from 35 were available for analysis. Details regarding biopsy sample processing, tissue genomic DNA library preparation, and sequencing have been described previously[8]. Peripheral blood samples were collected longitudinally before, during, and after CRT, whenever possible. Blood was collected in 10 mL EDTA vacutainer tubes and subsequent process was conducted within 4 h after collection. Blood samples were centrifuged firstly at $800 \times g$ for 15 min, followed by a second centrifugation at $1600 \times g$ for 10 min. Double-spun plasma was then separated and stored at −80 °C until ctDNA extraction was conducted. ctDNA was extracted from ≥1 mL of plasma using a QIAamp Circulating Nucleic Acid Kit (Qiagen). The QIAamp DNA FFPE Tissue kit and DNeasy Blood and Tissue kit (Qiagen, USA) were used to isolate genomic DNA from biopsy and whole blood samples. Extracted DNA was eluted in a volume ratio of 20 μL for every mL of plasma. The DNA samples were stored at −20 °C until analysis was performed.

### Library preparation, sequencing and mutation detection

Customized panel including 474 cancer-relevant genes were applied (Radiotron, Nanjing Geneseeq Technology, Inc.) (Supplementary Fig. 1). The mean coverage depth of the whole blood control was 190X, and 1600X for the tumor tissues samples. The mean coverage sequencing depth for the ctDNA samples was 5000X. Detailed information on the library preparation and sequencing was provided in supplementary methods of Supplementary File.

Trimmomatic was used for FASTQ file quality control by removing low-quality leading/trailing with readings below 20 bases or N bases. PCR duplicates were removed using Picard, and local realignment around insertions/deletions (indels) was performed. GATK3 was used for base quality score recalibration. Tumors and corresponding white blood samples were aligned and checked for SNP fingerprints using VCF2LR (GeneTalk). Unmatched samples and samples with a mean depth less than 30X in blood and less than 600X in plasma were excluded from further analyses. MuTect was used to detect somatic single nucleotide variants (SNV), and indels were detected using Scalpel (scalpel discovery in somatic mode). The criteria used for SNVs and indels filtration was provided in the supplementary methods of Supplementary File. Pathways analyses were performed using the oncogenic signaling pathways from The Cancer Genome Atlas and ImmPort databases[33,34].

### Determination positive detection of ctDNA and TMB

CtDNA-positive was defined as any detection of somatic variants that overlapped with tissue or level I/II variants annotated by the AMP/ASCO/CAP guide and OncoKB database in the ctDNA analyses[35]. Tissue TMB or bTMB was defined as the total number of missense mutations obtained by summing all base substitutions and indels in the coding region of targeted genes, which included synonymous alterations, but excluded driver mutations according to previous study[36].

### Statistical analysis

Oncoplots constructed using R (version 3.5.3) were used to visualize the overall mutation landscape. The final follow-up time was 1 November 2022. In survival analyses, Kaplan–Meier curves were compared using the log-rank test in R (version 4.1.2), and hazard ratios (HRs) with 95% confidence intervals (CIs) were calculated using the Cox proportional hazards model in R (version 4.1.2). Categorical variables were compared using the chi-square test or two-tailed Fisher's exact test. To evaluate the sensitivity, specificity, positive predictive value (PPV), and negative predictive value (NPV) of PET and ctDNA, we compared their predictive results with the final response determined by the gold standard assessment, and two-tailed Fisher's exact test was used to compare the difference. Statistical analyses were performed using the SPSS version 22.0 software (SPSS Inc., Chicago, IL, USA) and Stata software version 12.0. Statistical significance was set at $P < 0.05$, unless otherwise indicated.

### Reporting summary

Further information on research design is available in the Nature Portfolio Reporting Summary linked to this article.

## Data availability

The raw sequence data used in this study are available in the Genome Sequence Archive (Genomics, Proteomics & Bioinformatics 2021) in National Genomics Data Center (Nucleic Acids Res 2022), China National Center for Bioinformation/Beijing Institute of Genomics, Chinese Academy of Sciences database under accession code HRA005042. Requests are to be made to Dr. Mian Xi (ximian@sysucc.org.cn) describing the proposed research. Access can be obtained for academic use only under a data transfer agreement and upon Ethics Committee approval. The timescale for this process is approximately 4 months and the data will be available for 2 years. The OncoKB database used in this study is publicly available via https://www.oncokb.org/. The remaining data including individual de-identified participant data are available within the Source Data file. Study protocol was provided as Supplementary File. Source data are provided with this paper.

## Code availability

Oncoprints were applied to visualize multiple genetic alterations. The 'ComplexHeatmap' package provides the oncoPrint() function in

R (V 3.5.3). The 'survival' package in R (V4.1.2) was used to calculate probabilities and to plot KM curves. The code is accessible at https://github.com/b123r45678/R_code.git.

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

## Acknowledgements

This study was supported by grants from the National Natural Science Foundation of China (No.82172669 and 82373214: M.X.; No. 81972614 and 82273032: H.Y.), the Sci-Tech Project Foundation of Guangzhou (202201010933; Baoqing Chen), Beijing Xisike Clinical Oncology Research Foundation (Y-MSDZD2022-0878; M.X.), National Natural Science Foundation of Guangdong Province (2022A1515012483; SL.L.). The work of M.P.D. is supported by Berlin Institute of Health (Clinician Scientist Program) and by DKTK Berlin (Young Investigator Grant 2022). We thank all participating individuals. All authors give consent for the publication of the manuscript. We are grateful for the language editing services provided by ELSEVIER.

## Author contributions

Baoqing Chen, S.L. and M.X. conducted the conception and design of this study. Baoqing Chen, Yujia Zhu, Q.L., Y.H., H.Y., M.L. and M.X. provided study materials and patients. Baoqing Chen, S.L., Yujia Zhu, Q.L., Biqi Chen, Y.H., H.Y., M.L. and M.X. conducted data collection and assembly. Baoqing Chen, S.L., R.W., X.C., M.P.D., Yaru Zhang. and M.X. analyzed and interpreted the data. All authors were involved in the manuscript writing.

## Competing interests

Y.Z. is the employee of Nanjing Geneseeq Technology Inc. All other authors have no conflicts of interest to declare.

## Additional information

[1]State Key Laboratory of Oncology in South China, Guangdong Provincial Clinical Research Center for Cancer, Guangdong Esophageal Cancer Institute, Guangzhou, Guangdong, PR China. [2]Department of Radiation Oncology, Sun Yat-sen University Cancer Center, Guangzhou, Guangdong, PR China. [3]Institute of Pathology, Charité-Universitätsmedizin Berlin, corporate member of Freie Universität Berlin, Humboldt-Universität zu Berlin, Berlin, Germany. [4]Berlin Institute of Health, Berlin, Germany. [5]German Cancer Consortium (DKTK), Partner Site Berlin, German Cancer Research Center (DKFZ), Heidelberg, Germany. [6]Nanjing Geneseeq Technology Inc, Nanjing, Jiangsu, PR China. [7]Department of Thoracic Surgery, Sun Yat-sen University Cancer Center, Guangzhou, Guangdong, PR China. [8]These authors contributed equally: Baoqing Chen, Shiliang Liu, Yujia Zhu, Ruixi Wang. [9]These authors jointly supervised this work: Qiaoqiao Li, Hong Yang, Mian Xi. ✉e-mail: liqq@sysucc.org.cn; yanghong@sysucc.org.cn; ximian@sysucc.org.cn

