## [Peer Review File · Nature Communications]

Predictive role of ctDNA in esophageal squamous cell carcinoma receiving definitive chemoradiotherapy combined with toripalimabReviewers' Comments:

Reviewer #1:

Remarks to the Author:

In this study, Chen et al. explored the role of ctDNA and bTMB detection in monitoring the response and patient survival to CRT combined with immunotherapy in ESCC, based on a clinical trial which was run by the same team. In their phase II trial (n=42), Toripalimab combined with CRT achieved promising efficacy, which showed a cCR of 62% in locally advanced ESCC. Here, they showed that baseline ctDNA may serve as a biomarker of tumor burden. More importantly, they found that ctDNA-negative patients achieved a higher clinical complete response rate (cCR) and better survival than those with detectable ctDNA either during or post CRT. Patients with high bTMB levels during CRT also had prolonged OS.

My assessment of the work is: 1) it has novelty since ctDNA and bTMB detection has not hitherto been performed in the setting of CRT combined with immunotherapy in ESCC; 2) the finding of predictive and prognostic values of liquid biomarkers is very significant, given their promising utility. 3) overall the methodologies and data analyses are sound and result interpretation is appropriate.

I do have a number of concerns and questions, and I believe addressing these would help improve the quality and significance of the manuscript.

A major question needs to be clarified is the relationship between ctDNA positivity and bTMB levels, which I would assume that there is certain positive correlation given that both are surrogates representing fragmented tumor DNA released by tumor cells into the circulation. However, the authors showed an opposite pattern in terms of survival association. Specifically, patients with detectable ctDNA during CRT had shorter progression-free survival ($p=0.014$). Patients with post-CRT detectable ctDNA had a significantly shorter PFS ($p=0.012$) and worse overall survival (OS) ($p=0.004$). Yet, patients with high bTMB levels during CRT had prolonged OS ($p=0.027$). Can authors provide explanations to reconcile this seemingly opposite association patterns by exploring more on the relationship between ctDNA positivity and bTMB levels.

In Fig.1D, plasma and tissue shared 85 (58%) mutations, whereas 62 (42%) of the mutations were unique to plasma ctDNA analyses. Can authors explain why these 62 were missing in the tumor tissues? Could it be because the tumors were sequenced at much lower coverage? Also, have the authors performed any validation of these 62 mutations to ascertain if they are true mutations?

This is an important point, because the authors defined ctDNA-positive as any detection of somatic variants that overlapped with tissue or level I/II variants annotated by the OncoKB database. More importantly, ctDNA-positivity was related to treatment response, which is the key conclusion of the paper. Therefore, it is critical to ascertain the nature of these 62 mutations.

It is very important and standard practice that the sequencing data should be made immediately publicly available.

Ref #12 was cited as an esophageal cancer study but it was actually a lung cancer paper.

In the analysis of TMB, it was described that "Tissue TMB or bTMB was defined as the total number of missense mutations obtained by summing all base substitutions and indels in the coding region of targeted genes, which included synonymous alterations, but excluded driver mutations according to previous study.". I was wondering why driver mutations were excluded?

In table 1, the sensitivity and specificity of ctDNA and PET-CT in predicting cCR seems to be on par with each other. The authors performed statistical analysis to compare these two methods. Could authors explain in detail how the statistical analysis was done for the comparison, since I cannot find it in the manuscript.

Lastly, have the authors looked at the predictive value of tissue TMB in this patient cohort? Since tissue TMB is commonly used, it can serve as a benchmark to evaluate bTMB.

De-Chen Lin, PhD
University of Southern California

Reviewer #3:

Remarks to the Author:

Squamous cell carcinoma (SCC) is one of the most common types among histopathological types of esophageal cancer. For inoperable patients, standard care is considered definitive concurrent chemoradiotherapy. Several studies have been published recently depicting the effective role of several PD-1 or PD-L1 inhibitors in the treatment of ESCC to improve overall survival in combination with definitive concurrent chemoradiotherapy. This study is original and novel.

This study is also interesting and relevant. The authors showed the role of ctDNA and bTMB in monitoring the response and survival to CRT combined with immunotherapy in ESCC. The results illustrate the predictive value of ctDNA dynamic analyses and the use of bTMB to predict the clinical benefits for patients after anti-PD1 immunotherapy combined with CRT. It is interesting for the readers to know the prevalence of treatment methods and the factors in predictive outcomes.

However, I would show some flaws of this manuscript below:

1. The introduction part is quite complicated to understand the role of ctDNA, bTMB and ctDNA using next-generation sequencing (lines 87- 95). The authors should explain simpler and more concisely.
2. In material and method, which criteria for PET were used to assess treatment response? There are no specific criteria for PET. False positives due to inflammation after radiation therapy should be cautious.
3. Which test was used to compare the sensitivity and specificity of ctDNA and FDG PET/CT?
4. FDG PET/CT, CT, and endoscopy were used to assess treatment response after completion of CRT and what diagnostic tests were used further. As shown in Fig. 4 CT was used to follow-up. The authors should make clear the diagnostic modalities and the time point which were used.
5. This manuscript is quite complicated with many tables and figure which are hard to follow and track.

Reviewer #4:

Remarks to the Author:

NCOMMS-23-29018

Title: Predictive role of circulating tumor DNA and blood-based tumor mutational burden in esophageal squamous cell carcinoma receiving definitive chemoradiotherapy combined with toripalimab: Exploratory analyses of a phase II study (EC-CRT-001)

Chen et al., have presented an exploratory analysis of the utility of ctDNA and bTMB analysis as a biomarker for predicting response in patients with locally advanced esophageal squamous cell carcinoma (phase II study EC-CRT-001). The authors found that the presence of ctDNA in both the pre and post chemoradiation treatment (CRT) samples led to lower clinical complete response and poorer overall survival. Additionally, patients with higher bTMB during CRT had improved overall response.

While the authors note that ctDNA has been used to monitor patients who have undergone CRT for ESCC, the noteworthy result was the use of ctDNA for dynamic profiling during and post CRT. Also, the demonstration that ctDNA analysis could potentially be used to complement the PET-CT in predicting which patients may have radiation esophagitis was significant.

Further information to confirm the total cfDNA concentration detected for each patient, if unique molecular indexes were used during the cfDNA sequencing analysis, and a list of the mutations detected. A complete list of the variant allele frequencies for the baseline, during and post CRT ctDNA analysis would be extremely helpful, particularly in amplifications that are notoriously hard to detect in ctDNA analysis. It is not apparent whether these were detectable during or post-CRT samples. The authors mention that they generated NGS data from whole blood NGS, however they provided no description of how these results impacted the findings of the plasma ctDNA sequencing. It would be interesting to have a comment or a supplementary file showing any mutations that were excluded due to the presence of CHIP.

In line 214, the ctDNA concentration was stated as 26.3ng/mL. Can the author describe how they determined this concentration, or was this concentration from the entire amount of cell-free DNA detected in the extracted plasma sample?

Line 295 – description of PET-CT yielded a sensitivity of 79%, and a specificity of 90%. How was the status of cCR determined in the clinical setting?

I note that the authors concluded only patients with an NFE2L2 mutation had a negative correlation with PFS and OS. Were these mutations detected in more advanced patients? For some of the other mutations reported, such as EP300, the 95% CI is very broad. Can the authors comment on whether a larger cohort of patients may lead to statistical significance for other mutations?

In the publication, there is extensive use of Kaplan Meyer curves with labels of “number at risk” at the bottom of these curves. I found this confusing, could the authors more label?

Line 327 – define CPS (I assume it is combined positive score?)

In this publication, the author strongly presents the use of bTMB as a valid marker for treatment response. However, they have used bTMB analysis cutoff scores of 1, 2, 3, 4. Can they confirm that this is mutations/MB? I question how these cutoffs can accurately be used for the analysis, especially when the paper they cite (Kim et al.,) uses a pre-defined bTMB of >16 as high bTMB. I am not convinced that changing the bTMB threshold for the two different analyses is valid or whether this has been done to fit the data because of the higher number of mutations detected post-CRT. While the authors note in the discussion that the bTMB is a limitation of the study, I think many conclusions are drawn from this data. I also think the statement of line 434 or 435 is irrelevant because in the context of tumours with low ctDNA shedding, additional NGS will not likely achieve an accurate bTMB.

In summary, I believe this paper does extend the knowledge in this field, and with some additional clarification and supporting data, I believe this paper meets the standards for publication.

Reviewer #5:

Remarks to the Author:

In this manuscript, Chen, Liu, Zhu, Li, Yang, Xi, and colleagues explore the use of ctDNA and blood-based mutational burden to predict clinical complete response following definitive chemoradiotherapy with toripalimab in 42 patients as an exploratory arm of a clinical trial. Overall, while this study is well done with many clinical correlates, the n is quite small and even at these small numbers, the sensitivity and specificity of ctDNA (as measured in this study) as a test is too poor to be used as a clinical test.

Major comments:

1. How were the 474 cancer-relevant genes selected? Are these known to be highly mutated in esophageal SCC? Given that only 73% of patients had detectable ctDNA at baseline, it seems to indicate that the panel selected did not cover all the potential mutations. Given that there is matching

tumor mutational data, could some known mutations as identified from the tumor be sought using PCR from plasma?

2. The manuscript discusses predictive value, but while sensitivity and specificity is reported, the positive and negative predictive values are not.

3. I think that 2 patients is too few to say that ctDNA is an effective supplementary method to PET-CT.

4. The measurement of CR is important. CT PET and EGD were done. Was there random biopsy of the previous tumor bed?

Minor comments:

1. I think green might be a better color for CR in the figures?

Response letter:

We have carefully considered all the comments and suggestions and have made the necessary revisions accordingly. We believe that these revisions have significantly improved the quality and clarity of the manuscript. Changes to our manuscript were all highlighted within the document by using red colored text. Below, we provide our responses and the corresponding revisions made in the manuscript.

REVIEWER COMMENTS

Reviewer #1 (Remarks to the Author): expertise in ESCC genomics

In this study, Chen et al. explored the role of ctDNA and bTMB detection in monitoring the response and patient survival to CRT combined with immunotherapy in ESCC, based on a clinical trial which was run by the same team. In their phase II trial (n=42), Toripalimab combined with CRT achieved promising efficacy, which showed a cCR of 62% in locally advanced ESCC. Here, they showed that baseline ctDNA may serve as a biomarker of tumor burden. More importantly, they found that ctDNA-negative patients achieved a higher clinical complete response rate (cCR) and better survival than those with detectable ctDNA either during or post CRT. Patients with high bTMB levels during CRT also had prolonged OS.

My assessment of the work is: 1) it has novelty since ctDNA and bTMB detection has not hitherto been performed in the setting of CRT combined with immunotherapy in ESCC; 2) the finding of predictive and prognostic values of liquid biomarkers is very significant, given their promising utility. 3) overall the methodologies and data analyses are sound and result interpretation is appropriate.

I do have a number of concerns and questions, and I believe addressing these would help improve the quality and significance of the manuscript.

1: A major question needs to be clarified is the relationship between ctDNA positivity and bTMB levels, which I would assume that there is certain positive correlation given that both are surrogates representing fragmented tumor DNA released by tumor cells into the circulation. However, the authors showed an opposite pattern in terms of survival association. Specifically, patients with detectable ctDNA during CRT had shorter progression-free survival (p=0.014). Patients with post-CRT detectable ctDNA had a significantly shorter PFS (p=0.012) and worse overall survival (OS) (p=0.004). Yet, patients with high bTMB levels during CRT had prolonged OS (p=0.027). Can authors provide explanations to reconcile this seemingly opposite association patterns by exploring more on the relationship between ctDNA positivity and bTMB levels.

Reply: Thank you very much for your valuable comments. The different roles of ctDNA and bTMB in predicting survival have been extensively studied and reported.

Previous studies have demonstrated that non-small cell lung cancer (NSCLC) patients with ctDNA-positive status have a poor prognosis, while those with higher bTMB levels tend to exhibit a better response to immunotherapy and longer survival. (Wang Z et al. JAMA Oncol, 2019. PMID: 30816954; Kim ES et al. Nat Med, 2022. PMID: 35422531; Yang Y et al. Mol Cancer, 2022. PMID: 35590322; Pellini B et al. Clin Cancer Res, 2023. PMID: 37702716). Furthermore, our analysis revealed no significant difference in bTMB levels during or after CRT between ctDNA-negative and ctDNA-positive patients. bTMB levels were higher in ctDNA-positive patients at baseline, but neither pretreatment ctDNA nor bTMB levels could predict patient survival (as shown in the following table):

	Mean bTMB (mutations/mb)		P -value
	ctDNA-negative	ctDNA-positive	
Baseline	1.1	5.3	0.001
During-CRT	2.3	2.7	0.73
Post-CRT	2.8	3.2	0.82

The discrepancy in the definition of these two biomarkers may explain their irrelevance. ctDNA status serves as a qualitative indicator, categorized as either positive or negative, whereas bTMB is a quantitative indicator represented by an exact value. In this study, ctDNA-positive was defined as the detection of somatic variants overlapping with tissue or level I/II variants. On the other hand, bTMB was defined as the total number of missense mutations obtained by summing all base substitutions and indels in the coding region of the target genes, including synonymous changes (Chalmers ZR et al. Genome Med. 2017. PMID: 28420421). Notably, synonymous mutations are included in the calculation of bTMB, whereas ctDNA may not necessarily be classified as positive if synonymous mutations do not overlap with the level I/II variants.

2: In Fig.1D, plasma and tissue shared 85 (58%) mutations, whereas 62 (42%) of the mutations were unique to plasma ctDNA analyses. Can authors explain why these 62 were missing in the tumor tissues? Could it be because the tumors were sequenced at much lower coverage? Also, have the authors performed any validation of these 62 mutations to ascertain if they are true mutations? This is an important point, because the authors defined ctDNA-positive as any detection of somatic variants that overlapped with tissue or level I/II variants annotated by the OncoKB database. More importantly, ctDNA-positivity was related to treatment response, which is the key conclusion of the paper. Therefore, it is critical to ascertain the nature of these 62 mutations.

Reply: We appreciate the reviewer's comments. The discrepancy between ctDNA and tumor tissue sequencing results could be attributed to several reasons, such as the limited representation of spatial and temporal heterogeneity in tumor tissue sequencing or the differences in sequencing depth between tissue and blood-based NGS methods. Mutation variants discordance between matched ctDNA and tumor samples has been widely documented in various cancer types. For instance, in NSCLC, the NGS-detected

mutation concordance rate between tissue and plasma samples was approximately 60%, with substantial somatic variants identified exclusively in ctDNA compared to the tissue samples (Metzenmacher M et al. *Transl Oncol*, 2022. PMID: 34800919; Shu Y et al. *Sci Rep*, 2017. PMID: 28373672). Similarly, in gastroesophageal adenocarcinoma, discordance rates ranging from 60% to 87% have been reported (Maron SB et al. *Clin Cancer Res*, 2019. PMID: 31427281; Kato S et al. *Clin Cancer Res*, 2018. PMID: 30348637).

We implemented stringent steps in our ctDNA analysis pipeline to ensure the accuracy of the detected mutations. First, all variants identified in tumor and ctDNA analyses underwent rigorous filtering using strict criteria, as described in the Supplementary methods section outlined in additional file 1. Second, ctDNA variants that were not present in the corresponding primary tumor were rigorously evaluated and were considered true variants only if all of the following criteria were met: (i) alter reads ≥ 3 ; (ii) absence in the internal database of clonal hematopoiesis variants obtained from peripheral blood leukocyte samples of approximately 500 healthy donors; (iii) variant allele frequency (VAF) = 0% in the paired peripheral blood leukocyte control sample, thereby excluding clonal hematopoiesis variants or somatic mutations. Detailed information on these criteria for ctDNA analysis can be found in the Supplementary methods section of Additional file 1.

We fully agree that further validation by PCR would be a valuable addition to our results. Alternative PCR methods, such as digital PCR (dPCR) or Amplification Refractory Mutation System PCR (ARMS-PCR) have been used for a specific mutation detection in ctDNA analyses. However, we regret that multiple individual PCR reactions are not feasible in this cohort because all available DNA was used for NGS sequencing. Multiple PCR reactions require a substantial amount of DNA, which is a significant challenge when processing peripheral blood samples. In addition, the development of different reaction systems for this purpose is time-consuming and inconvenient, especially in the context of clinical practice.

3. It is very important and standard practice that the sequencing data should be made immediately publicly available.

Reply: We apologize for the delay in making the primary data accessible in the GSA Human database. It is now available with the accession number: HRA005042 (<https://ngdc.cncb.ac.cn/gsa-human/browse/HRA005042>).

4. Ref #12 was cited as an esophageal cancer study but it was actually a lung cancer paper.

Reply: Thank you for highlighting this point. We mentioned that “studies focusing on the detection of MRD by ctDNA after completion of definitive CRT with curative intent have also been performed for **locally advanced malignancies**, including esophageal

carcinoma”. To avoid potential misunderstanding, we have changed this sentence in the revised paper to “studies focusing on the detection of MRD by ctDNA after completion of definitive CRT with curative intent have also been performed for **lung cancer** and esophageal carcinoma” (line 86-87).

5. In the analysis of TMB, it was described that “Tissue TMB or bTMB was defined as the total number of missense mutations obtained by summing all base substitutions and indels in the coding region of targeted genes, which included synonymous alterations, but excluded driver mutations according to previous study.” I was wondering why driver mutations were excluded?

Reply: We appreciate the reviewer's comments. Including or excluding driver mutations in the calculation of tumor mutation burden (TMB) during NGS has not yielded conclusive results. However, filtering out driver mutations may help mitigate the effect of overrepresented cancer genes that may artificially inflate TMB counts. Furthermore, it can reduce the influence of different gene compositions across various panels (Doig KD et al. Pathology, 2022. PMID: 35153070). Most of FDA-approved commercial tumor NGS panels, such as FoundationOne CDx, Omics Core, FoundationOne Liquid CDx, and PGDx Elio Tissue Complete, have consistently adopted the approach of excluding driver mutations in TMB calculations. In our study, we also employed this calculation method, excluding driver mutations.

In addition, previous research has demonstrated a strong correlation between TMB quantified using this approach in our gene panel and whole exome sequencing (WES) results, which further supporting the reliability and suitability of this TMB calculation method (Fang W et al. Clin Cancer Res, 2019. PMID: 31085721; Chalmers ZR et al. Genome Med, 2017. PMID: 28420421).

6. In table 1, the sensitivity and specificity of ctDNA and PET-CT in predicting cCR seems to be on par with each other. The authors performed statistical analysis to compare these two methods. Could authors explain in detail how the statistical analysis was done for the comparison, since I cannot find it in the manuscript.

Reply: Thank you for your reminder. The assessment of treatment response was conducted independently by two experienced radiologists using PET imaging. Complete response (CR) on PET was defined as either the maximum standardized uptake value (SUVmax) in the primary region being at a normal physiological level or, if SUVmax was higher than normal, it was evenly distributed and overlapped with the radiation field, indicating esophagitis (Suzuki A et al. Ann Oncol, 2013. PMID: 23994746; Suzuki A et al. Cancer, 2011. PMID: 21456015).

The gold standard assessment of clinical response included comprehensive methods: esophagogastroduodenoscopy, CT, PET-CT, and bite-on-bite biopsy. To evaluate the sensitivity and specificity of PET and ctDNA, we compared their results with the final

response determined by the gold standard assessment respectively. We used a two-tailed Fisher's exact test to compare the sensitivity and specificity between the two methods. We have added this information in the Methods section of the revised paper (lines 177-180).

7. Lastly, have the authors looked at the predictive value of tissue TMB in this patient cohort? Since tissue TMB is commonly used, it can serve as a benchmark to evaluate bTMB.

Reply: We appreciate your suggestions. We had analyzed and found no significant prognostic role of tissue TMB in predicting the survival in this cohort. This result was previously reported in our report of the primary results of this clinical trial (Zhu Y et al. *Lancet Oncol*, 2023. PMID: 36990609).

Reviewer #3 (Remarks to the Author): expert in PET-CT imaging in ESCC

Squamous cell carcinoma (SCC) is one of the most common types among histopathological types of esophageal cancer. For inoperable patients, standard care is considered definitive concurrent chemoradiotherapy. Several studies have been published recently depicting the effective role of several PD-1 or PD-L1 inhibitors in the treatment of ESCC to improve overall survival in combination with definitive concurrent chemoradiotherapy. This study is original and novel.

This study is also interesting and relevant. The authors showed the role of ctDNA and bTMB in monitoring the response and survival to CRT combined with immunotherapy in ESCC. The results illustrate the predictive value of ctDNA dynamic analyses and the use of bTMB to predict the clinical benefits for patients after anti-PD1 immunotherapy combined with CRT. It is interesting for the readers to know the prevalence of treatment methods and the factors in predictive outcomes.

However, I would show some flaws of this manuscript below:

1. The introduction part is quite complicated to understand the role of ctDNA, bTMB and ctDNA using next-generation sequencing (lines 87- 95). The authors should explain simpler and more concisely.

Reply: We appreciate your critical comments, and we have addressed this issue by revising and condensing the relevant section in the revised paper (lines 87-97).

2. In material and method, which criteria for PET were used to assess treatment response? There are no specific criteria for PET. False positives due to inflammation after radiation therapy should be cautious.

Reply: Thank you for your insightful comments. We fully agree with the need for caution regarding false positives caused by inflammation following radiation therapy.

To ensure the specificity of the PET/CT assessment, the PET/CT-evaluated response was independently assessed by two experienced radiologists. In cases of disagreement, a third radiologist was involved to determine the final outcome. A complete response (CR) determined by PET was defined when the maximum standardized uptake value (SUV_{max}) in the primary region was at a normal physiological level or when SUV_{max} was higher than normal but exhibited a uniform distribution and overlapped with the radiation field, indicating esophagitis (Suzuki A et al. *Ann Oncol*, 2013. PMID: 23994746; Suzuki A et al. *Cancer*, 2011 PMID: 21456015). We have added this information in the Methods section of the revised paper (lines 121-124).

3. Which test was used to compare the sensitivity and specificity of ctDNA and FDG PET/CT?

Reply: Thank you for the query. To evaluate the sensitivity and specificity of PET and ctDNA, we compared their results with the definitive response determined by the gold standard assessment. The gold standard assessment involved comprehensive methods: esophagogastroduodenoscopy, CT, PET-CT, and bite-on-bite biopsy. We used a two-sided Fisher's exact test to compare the sensitivity and specificity between the two methods. This additional information has been added in the Methods section (lines 177-180).

4. FDG PET/CT, CT, and endoscopy were used to assess treatment response after completion of CRT and what diagnostic tests were used further. As shown in Fig. 4 CT was used to follow-up. The authors should make clear the diagnostic modalities and the time point which were used.

Reply: Thank you for raising this point. According to the protocol of this clinical trial (Zhu Y et al. *Lancet Oncol*, 2023. PMID: 36990609), all patients were staged or assessed for response using PET/CT, CT, and esophagoscopy at the time of diagnosis and three months after completion of radiotherapy. During the follow-up period, patients were scheduled for regular visits every 9 weeks for the first year, every 3 months for the second year, and then every 6 months thereafter. Follow-up included physical examinations, serological tests, esophageal barium X-ray, and enhanced CT scans of the chest and abdomen.

During the first three years, esophagoscopy was performed every six months, and pathological biopsies were taken if suspicious lesions were found. PET-CT scans were required for suspicious recurrence or metastatic lesions that could not be determined by CT during regular follow-up. In the case of patient #17 shown in Figure 4, no obvious lesions or significant signals were observed at later follow-ups, so routine PET-CT scans were not performed. This information about the diagnostic modalities and the follow-up time point has been included in the Supplementary methods section of the revised paper (Additional file 1, page 2-3).

5. This manuscript is quite complicated with many tables and figure which are hard to follow and track.

Reply: Thank you for your critical comment regarding the complexity of the manuscript due to the numerous tables and figures. We recognize the importance of clarity and ease of navigation for the reader. We have made revisions to improve the description and interpretation of the results, aiming to provide clearer and more accurate information.

Reviewer #4 (Remarks to the Author): expertise in longitudinal ctDNA genomics analysis

NCOMMS-23-29018

Title: Predictive role of circulating tumor DNA and blood-based tumor mutational burden in esophageal squamous cell carcinoma receiving definitive chemoradiotherapy combined with toripalimab: Exploratory analyses of a phase II study (EC-CRT-001).

Chen et al., have presented an exploratory analysis of the utility of ctDNA and bTMB analysis as a biomarker for predicting response in patients with locally advanced esophageal squamous cell carcinoma (phase II study EC-CRT-001). The authors found that the presence of ctDNA in both the pre and post chemoradiation treatment (CRT) samples led to lower clinical complete response and poorer overall survival. Additionally, patients with higher bTMB during CRT had improved overall response.

While the authors note that ctDNA has been used to monitor patients who have undergone CRT for ESCC, the noteworthy result was the use of ctDNA for dynamic profiling during and post CRT. Also, the demonstration that ctDNA analysis could potentially be used to complement the PET-CT in predicting which patients may have radiation esophagitis was significant.

1. Further information to confirm the total cfDNA concentration detected for each patient, if unique molecular indexes were used during the cfDNA sequencing analysis, and a list of the mutations detected. A complete list of the variant allele frequencies for the baseline, during and post CRT ctDNA analysis would be extremely helpful, particularly in amplifications that are notoriously hard to detect in ctDNA analysis. It is not apparent whether these were detectable during or post-CRT samples.

Reply: Thank you for this valuable point. A list of all the mutations and the corresponding variant allele frequencies at tumor tissue, baseline-, during-, and post-CRT ctDNA analysis is provided in Additional file 2. Consistent with other studies of ctDNA, we observed rare instances of amplification in the ctDNA analyses. Specifically, we found amplification variants in only six patients at baseline, whereas no detectable amplification variants were identified in the during- or post-CRT samples.

2. The authors mention that they generated NGS data from whole blood NGS, however they provided no description of how these results impacted the findings of the plasma ctDNA sequencing. It would be interesting to have a comment or a supplementary file showing any mutations that were excluded due to the presence of CHIP.

Reply: All variants in tumor and ctDNA analyses were filtered with strict criteria to exclude the non-specific somatic variants, including clonal hematopoiesis variants (CHIP). The following criteria were applied: (i) alter reads ≥ 3 ; (ii) not present in the internal clonal hematopoiesis variants database of normal using peripheral blood leukocyte samples from ~500 healthy donors; (iii) VAF = 0% in the paired peripheral blood leukocyte control sample to further exclude the clonal hematopoiesis variants or germline mutation. We had added the information on the criteria for ctDNA analyses in the Supplementary Method in additional file 1 (page 2, paragraph 2).

The list of the mutations that were excluded from the analysis due to the presence of clonal hematopoiesis is shown below:

Sample	Gene	AA Change	Exonic Func	AF
P1_S3	TET2	c.1648C>T(p.R550*)	stop_gained	0.59%
P2_S3	TET2	c.1648C>T(p.R550*)	stop_gained	0.38%
P1_S6	TET2	c.3889G>A(p.G1297R)	missense_variant	0.43%
P2_S6	TET2	c.3889G>A(p.G1297R)	missense_variant	1.04%
P3_S6	TET2	c.3889G>A(p.G1297R)	missense_variant	1.39%
P1_S19	DNMT3A	c.2347A>T(p.K783*)	stop_gained	0.26%
P2_S19	DNMT3A	c.2347A>T(p.K783*)	stop_gained	0.20%
P3_S19	DNMT3A	c.2347A>T(p.K783*)	stop_gained	0.54%
P1_S24	DNMT3A	c.2645G>A(p.R882H)	missense_variant	0.28%
P2_S24	DNMT3A	c.2645G>A(p.R882H)	missense_variant	0.24%
P3_S24	DNMT3A	c.2645G>A(p.R882H)	missense_variant	2.39%
P1_S38	DNMT3A	c.941G>A(p.W314*)	stop_gained	1.11%
P1_S38	DNMT3A	c.1826del(p.F609Sfs*42)	frameshift_variant	1.11%
P2_S38	DNMT3A	c.1826del(p.F609Sfs*42)	frameshift_variant	1.11%
P2_S38	DNMT3A	c.941G>A(p.W314*)	stop_gained	0.84%
P3_S38	DNMT3A	c.941G>A(p.W314*)	stop_gained	1.50%
P3_S38	DNMT3A	c.1826del(p.F609Sfs*42)	frameshift_variant	1.47%
*P1: preCRT ctDNA; P2: duringCRT ctDNA; P3: postCRT ctDNA; S: Sample ID; AF: Allele Frequency;				

3. In line 214, the ctDNA concentration was stated as 26.3ng/mL. Can the author describe how they determined this concentration, or was this concentration from the entire amount of cell-free DNA detected in the extracted plasma sample?

Reply: We used Qubit fluorometer assays to measure the DNA concentration. The concentration of ctDNA that we mentioned in the manuscript is the concentration of the total amount of cell-free DNA detected in the extracted plasma sample. We apologize for this inaccurate description and have revised it to ‘concentration of cell-free DNA’ (line 223).

4. Line 295 – description of PET-CT yielded a sensitivity of 79%, and a specificity of 90%. How was the status of cCR determined in the clinical setting?

Reply: In this study, we defined complete response (CR) based on comprehensive methods using the following criteria: 1) absence of any identifiable lesion, budding, or ulceration on esophagogastroduodenoscopy (EGD); 2) negative endoscopic findings confirmed by bite-on-bite biopsy; 3) no evidence of local or distant recurrence and a maximum standardized uptake value (SUVmax) in the primary region at a normal physiological level or distributed in an esophagitis pattern according to PET imaging. Whenever there was any uncertainty in the findings, a re-evaluation was performed within 6 weeks to determine the final response. These details have been previously described in our report on the primary results of the clinical trial (Zhu Y et al. Lancet Oncol, 2023. PMID: 36990609). We also attached this information in the Supplementary Method in additional file 1 (page 2-3).

5. I note that the authors concluded only patients with an *NFE2L2* mutation had a negative correlation with PFS and OS. Were these mutations detected in more advanced patients? For some of the other mutations reported, such as EP300, the 95% CI is very broad. Can the authors comment on whether a larger cohort of patients may lead to statistical significance for other mutations?

Reply: In this cohort, only three patients were classified as stage I-II, while 23 patients were classified as stage III, and the remaining 14 patients were classified as stage IVA. Among the pre-CRT ctDNA-positive patients, all were diagnosed with stage III or IV disease. We found that the *NFE2L2* mutation was more frequently observed in stage IV disease compared to stage III, although this difference did not reach statistical significance (8% vs. 25%, $p=0.17$). However, it should be noted that our sample size was limited, and it is difficult to draw definitive conclusions regarding the higher frequency of these mutations in more advanced stages. Interestingly, analyses based on an integrated dataset of 1930 ESCC genomic samples have suggested that common mutational variants, including *NFE2L2*, have a similar mutation rate across different stages of the disease (Li M et al. Nat Commun, 2022. PMID: 36071046).

We acknowledge the potential for achieving statistical significance for other mutations with a larger cohort of patients, along with narrower 95% confidence intervals. However, the lack of an external cohort for ctDNA analysis hinders our ability to validate these findings. In our exploration of an external tissue genomic dataset of ESCC, we observed similar results, specifically that only the *NFE2L2* mutation showed

a negative correlation with overall survival in ESCC (Table below; p=0.0416) (Song Y et al. Nature, 2014. PMID: 24670651). Several studies with larger case numbers have reported a high prevalence of *TP53*, *CDKN2A*, *NOTCH1*, and *EP300* mutations, but the correlation of these mutations with survival has yielded inconsistent results (Li M et al. Nat Commun, 2022. PMID: 36071046; Gao YB et al. Nat Genet, 2014. PMID: 25151357).

Gene	HR (95% CI)	Log-rank p-value
NFE2L2	2.80 (1.00~7.89)	0.0416
TP53	1.17 (0.49~2.78)	0.725
CDKN2A	0.40 (0.05~2.89)	0.344
LRP1B	1.80 (0.80~4.08)	0.152
NOTCH1	1.64 (0.58~4.61)	0.346
EP300	0.75 (0.10~5.48)	0.778

6. In the publication, there is extensive use of Kaplan Meyer curves with labels of “number at risk” at the bottom of these curves. I found this confusing, could the authors more label?

Reply: The “number at risk” represents the number of patients who remain under observation or at risk during the follow-up period.

7. Line 327 – define CPS (I assume it is combined positive score?)

Reply: Thank you for mentioning this. We had defined CPS (combined positive score) and added this point in the manuscript (line 336).

8. In this publication, the author strongly presents the use of bTMB as a valid marker for treatment response. However, they have used bTMB analysis cutoff scores of 1, 2, 3, 4. Can they confirm that this is mutations/MB? I question how these cutoffs can accurately be used for the analysis, especially when the paper they cite (Kim et al.,) uses a pre-defined bTMB of >16 as high bTMB. I am not convinced that changing the bTMB threshold for the two different analyses is valid or whether this has been done to fit the data because of the higher number of mutations detected post-CRT. While the authors note in the discussion that the bTMB is a limitation of the study, I think many conclusions are drawn from this data.

Reply: Thank you for this comment. The unit we used in the bTMB analyses is mutations/MB. It is important to note that the cut-off for tissue TMB in predicting response to immunotherapy varies across studies and depends on many factors such as cancer type, tumor stage, and treatment approaches.

The commonly reported cut-off value of 16 for high TMB has been observed primarily in studies of lung cancer, where TMB tends to be higher than in other cancers (Li L et al. Comput Struct Biotechnol J, 2021. PMID: 34745455). However, in a report involving 1145 ESCC patients, the median tissue TMB value was found to be 3.58,

which is significantly lower than that observed in melanoma and lung cancer (Zou B et al. *Front Mol Biosci*, 2022. PMID: 35127817). In our study post- or during-CRT bTMB values are similar and have a lower median value of 1 (IQR, 1–3). Additionally, the TMB of tumors with localized stages was lower compared to metastatic disease. The fact that all patients enrolled were in the locally advanced stage may contribute to the relatively low bTMB levels seen in this study.

Furthermore, it is widely recognized that the commonly defined high TMB cut-off value should not be applied universally across different cancer types or stages. In our study, we determined the cut-off based on the distribution of TMB values within the entire cohort. The results revealed a correlation between higher TMB and improved survival using two cut-offs in ctDNA analyses during- or post-CRT. However, our focus is not on the specific cut-off value, but rather on highlighting the trend of the correlation between high TMB and improved survival. It is important to acknowledge that the conclusions drawn from this small cohort may lack the statistical power to make definitive findings. However, our results do provide valuable insights suggesting that higher bTMB may serve as a potential biomarker for predicting survival outcomes. Further research with larger cohorts is warranted to validate these findings. Overall, it is important to consider the context of different cancer types, stages, and treatment modalities when interpreting TMB values and defining the optimal cut-offs for clinical practice. We had added these perspectives in the Discussion section (line 415-422).

I also think the statement of line 434 or 435 is irrelevant because in the context of tumours with low ctDNA shedding, additional NGS will not likely achieve an accurate bTMB. In summary, I believe this paper does extend the knowledge in this field, and with some additional clarification and supporting data, I believe this paper meets the standards for publication.

Reply: We apologize for the confusing interpretation. There have been conflicting results regarding the relationship between sequencing depth and TMB estimation (Makrooni MA et al. 2022. PMID: 35918650). Therefore, we have deleted this sentence in the revised paper according to your comments.

Reviewer #5 (Remarks to the Author): clinical expertise in ESCC

In this manuscript, Chen, Liu, Zhu, Li, Yang, Xi, and colleagues explore the use of ctDNA and blood-based mutational burden to predict clinical complete response following definitive chemoradiotherapy with toripalimab in 42 patients as an exploratory arm of a clinical trial. Overall, while this study is well done with many clinical correlates, the n is quite small and even at these small numbers, the sensitivity and specificity of ctDNA (as measured in this study) as a test is too poor to be used as a clinical test.

Major comments:

1. How were the 474 cancer-relevant genes selected? Are these known to be highly mutated in esophageal SCC? Given that only 73% of patients had detectable ctDNA at baseline, it seems to indicate that the panel selected did not cover all the potential mutations. Given that there is matching tumor mutational data, could some known mutations as identified from the tumor be sought using PCR from plasma?

Reply: We appreciate the reviewer's comment. The complete list of the 474 genes used in this panel can be found in the additional file Figure S1. This panel includes genes that are known to be associated with the response to chemoradiotherapy, targeted therapy, and immunotherapy in different cancers. The ctDNA status detected by this panel has been validated as an ideal biomarker for stratifying prognosis after chemoradiation in ESCC (Wang X et al. Clin Transl Med, 2022. PMID: 36437506).

Furthermore, this panel includes the majority of the top 20 genomic alterations (such as *TP53*, *RBI*, *CDKN2A*, *PIK3CA*, *NOTCH1*, *NFE2L2*, *EP3000*, *KEAP1*, *TGFBR2*, *CUL3*, etc.) reported in esophageal squamous cell carcinoma (Li M et al. Nat Commun, 2022. PMID: 36071046). Giving the lack of an FDA-approved NGS panel specifically for ESCC, this panel represents a practical choice for comprehensive genomic analysis in this context.

The ctDNA status may vary depending on the type and stage of the cancer. In localized disease, the ctDNA positive rate is generally lower than in metastatic disease due to the limited release of tumor DNA into the bloodstream. Previous studies have suggested ctDNA positive rates of approximately 50-70% in localized ESCC, and our reported rate of 73% is consistent with these findings (Huang Q et al. BMC Cancer, 2019. PMID: 31429737; Wang X et al. Clin Transl Med, 2022. PMID: 36437506; Ng HY et al. JAMA Surg, 2023. PMID: 37728901).

We acknowledge that further validation by PCR may be an option. However, unlike *EGFR* in lung cancer, there is no specific targetable mutation in ESCC, so ctDNA analysis would require multiple individual PCR reactions targeting different variants. This approach requires a significant amount of DNA, which is a major challenge when working with peripheral blood samples. Additionally, the development of different reaction systems for this purpose is time consuming and laborious, especially in the context of clinical practice.

2. The manuscript discusses predictive value, but while sensitivity and specificity is reported, the positive and negative predictive values are not.

Reply: Thank you for your valuable comment. We have added the positive and negative predictive values to Table 1.

3. I think that 2 patients are too few to say that ctDNA is an effective supplementary method to PET-CT.

Reply: Thank you for your valuable comments. We conducted a comparison of the sensitivity and specificity of PET-CT and ctDNA in predicting complete clinical response (cCR) based on the available post-treatment ctDNA analyses and PET-CT data from 37 patients. Due to the limited number of patients enrolled, the results should be interpreted with caution. We acknowledge that our study may lack sufficient statistical power to definitively conclude on the role of ctDNA as an adjunct to PET-CT. To address this limitation and provide a more comprehensive discussion, we have rephrased and expanded our interpretation of these findings in the Discussion section (lines 439-443).

4. The measurement of CR is important. CT PET and EGD were done. Was there random biopsy of the previous tumor bed?

Reply: Thank you for your comment. We performed EGD with bite-on-bite biopsies on the previous tumor bed according to the methodology described in the preSANO trial to assess tumor response (Noordman BJ et al. Lancet Oncol, 2018. PMID: 29861116).

In this study, we defined complete response (CR) based on comprehensive methods and the following criteria: 1) absence of any identifiable lesion, budding, or ulceration on esophagogastroduodenoscopy (EGD); 2) negative endoscopic findings confirmed by bite-on-bite biopsy; 3) no evidence of local or distant recurrence, and a maximum standardized uptake value (SUVmax) in the primary region at a normal physiological level or distributed in an esophagitis pattern according to PET imaging. If there was any uncertainty in the findings, re-evaluation was performed within 6 weeks to determine the final response. These details were previously described in our report of the primary results of this clinical trial (Zhu Y et al. Lancet Oncol, 2023. PMID: 36990609).

Minor comments:

1. I think green might be a better color for CR in the figures?

Reply: Thank you for your comment. We had changed the color of Figure 2D and Figure S8 based on your suggestion.

Reviewers' Comments:

Reviewer #1:

Remarks to the Author:

The authors have addressed all my concerns. I have no more comments for the revised manuscript. I appreciate the authors effort and responsiveness.

Reviewer #3:

Remarks to the Author:

The revised manuscript seems to be satisfied.

Reviewer #4:

Remarks to the Author:

I would like to thank the authors for addressing my comments. This has been done in a thorough and thoughtful manner. I am very pleased to confirm that I am happy for this paper to be published.

Reviewer #5:

Remarks to the Author:

The authors have made improvements to the manuscript and addressed my concerns.